# Application of Biographical Data in Student's Major Selection

**Yuting Wang \* and Guandong Song**

School of Humanities & Law, Northeastern University, No. 3-11, Wenhua Road, Heping District, Shenyang 110819, China

\* Correspondence: 1810016@stu.neu.edu.cn

**Abstract:** The research studies describe that students utilize Information and Communication Technology (ICT) widely to improve their academic performance. In the classroom, students use ICT assistive technologies via laptops and smartphones for academic and non-academic activities. The ICT tool interactions are applied to developing an effective learning environment that is used to support the student's learning and understanding in a specific context. The utilization of ICT motivates the students to utilize the technologies in the classroom environment. The ICT training policies help resolve the fundamental issues that students come across, particularly high school students going to college. However, most students do not know enough about their major tendencies and feel lost when deciding on a major. Our study aimed to apply ICT biographical data as a tool for major selection. Based on the rationale of psychometrics and valuable evidence, some studies show that the average high school score is the best predictor of the average college score. The biographical data prediction method is the pre-university life history of students of different majors. Compiling questionnaires takes the college academic performance of students as the studying criterion and weights projects on the biographical data table to develop college students' biographical information blank and its norm system to provide services for student's choice of major. Various results show that biographical information blank items are diverse, and the impurity of the content may lead to low internal reliability ($\alpha$ coefficient is usually between 0.60 and 0.80) but a high test–retest validity coefficient (usually between 0 and 0.90). In contrast, its validity has predictive validity because it is independent of each score. Furthermore, since biographical information blanks comprise verifiable and unverifiable items, the ideal subjects answered more reliably because they were accountable for their responses. Studies show that the description of individual life history was moderately associated with the results recorded by the psychologist.

**Keywords:** Information Communication Technology; assistive technology; ICT training policies; biographical data; biographical information blank; major selection; college student; prediction tools

## 1. Introduction

The development of Information and Communication Technologies (ICT) [1] is widely utilized in different social, health, and economic applications to make continuous information transformation. Education requires open systems and new resources to improve student learning abilities. In addition, the learning process requires a lot of resources to minimize the barriers while providing effective education. The ICT environment achieves quality and equity criteria in education systems because it supports various technology that helps process people's requests with minimum computation difficulties [2]. When discussing ICTs in sustainable learning environments, it is important to consider how they might be used to better the lives of historically marginalized populations, such as women, members of linguistic and racial minorities, the elderly, those with impairments, etc. The interaction between ICTs used in training contexts should help create learning environments that consider students' complete diversity, improve other teaching and learning methods, foster communal living and relationships, and support educational innovation focused on

equity. ICTs provide various opportunities, both in virtual and technology-supported face-to-face teaching situations, which can overcome the drawbacks of traditional educational systems and create learning environments. In-depth examination and discussion of ICTs as effective learning environments are the goals of this monograph. To do this, it is necessary to consider technology as a tool of approach for all individuals.

ICT assistive technologies [3] are widely utilized by students in the form of tablets, smartphones, and laptops because most students bring their smartphones and laptop to class. These devices use ICT, which greatly impacts classroom activities such as note taking and the student learning process. In addition, ICT is applied in the student's non-academic activities such as instant messaging, online shopping, browsing their accounts, and virtual gameplay. According to various research studies, 42% of students utilize ICT assistive technologies for their non-academic activities [4] via a laptop and mobile phone. Most studies indicate that ICT is used in the classroom by restricting non-academic activities. During this process, the student uses the ICT to record the sessions and online notes by considering the education rules and policies. The institutions are limited to student non-academic performance to minimize distractions during class time.

Most institutions utilize ICT assistive technologies to clarify student doubts and help make decisions in their higher education. The students use ICT-enabled devices to gather information regarding the lecture and easily access the materials. According to the search results, students can select their major subjects with minimum difficulties. In most situations, students find it difficult to understand the concept and confuse their learning terms, which is resolved using the ICT-provided supplemental information. Therefore, the ICT assistive technologies successfully provide supplementary information that helps students engage in learning activities.

Disputing the choice of major is a fundamental issue that students encounter, especially high school students going to college, because it plays a crucial role in future life choices. However, most students do not know enough about their major tendencies and feel lost in their major decisions. Many students realize their interests and talent are inconsistent with their current major. According to a previous social survey conducted by China Youth Daily, only 13.6% said they knew their chosen major well when applying to universities. At the same time, the rest were blind during the admission process [5]. In [6], clinical rotation called internship in business is discussed to have the option of enhancing overall performance. The internship process gives a temporary job which is relevant to the career or academic interest. In addition, it increases the professional role experience, screening the potential employees, while the talent acquisition team assists and updates the company information. This work uses several clinical internships, and these are transformative for students to enhance overall academic ideas and education growth.

In practical learning, only 16.0% felt that their major was in line with their expectations, 56.2% thought it did not fit, and 27.8% wanted to transfer to another major. In a 2018 college student major satisfaction survey, only 35% of college students were satisfied with their major, 12% were dissatisfied, and 53% were reluctant with their primary choice [5]. Certain students were incompatible with their current major selection, which is difficult to transfer, hurting their education.

To resolve the challenging issues for college-bound students to take the college entrance examination, some scholars have carried out pertinent investigations on additional tools via the Information and Communication Technology (ICT) for the college entrance examination, such as the self-efficacy scale for high school students compiled by Peng and Long (2001) [7]. Peng (2017) et al. [8] proposed a volunteer-assisted filling system based on intelligent decision making.

Furthermore, Schmidt et al. (2004) revealed that college student biographical data and situational judgment inventory have better predictive validity for future performance compared to college enrollment scores (SAT/ACT, etc.) [9]. Another study by Schmidt et al. (2007) showed that biographical data had predictive validity for foretelling college performance, absenteeism and withdrawal, and future development [10]. Reed et al.

established that biographical data have predictive rationality for university performance in academic history and personal perseverance dimensions [11]. Roenicke (2013) revealed that biographical data have good prognostic sincerity for college performance [12]. In addition, Kunsel et al. (2020) conducted a meta-analysis on the relationship between biographical data and student achievement. They showed that biographical data have apparent predictive effects on students' academic achievement, absenteeism, personal perseverance, and other aspects [13]. Baguan et al. (2019) [14] integrate Information and Communication Technology (ICT) in the Mauritius Higher education system to select the female teaching staff. The academic utilizes fewer ICT tools to increase the education system's performance. The system intends to examine the relationship between the female staff teaching activities and the ICT integration timeline factor. Then, thinking methodology is utilized to investigate the staff performance. The ICT technique determines staff efficiency according to the education system policy. In a nutshell, as an extrapolation tool for scientific talents, biographical data have the advantages of easy operation, considerable information, high effectiveness, and robust application. It has a good likelihood effect on students' future development orientation and can help students understand their major tendency and properly assess the trend of their primary choice. Abbas et al., 2021 [15] discuss the factors affected by students while selecting the subjects in their higher education in UK and Germany. During this process, six themes such as career, academic, financial, marketing, personnel, and social-related information are utilized. According to these characteristics, empirical analysis is performed to identify their subject selection in the UK and Germany. Vietze et al., 2022 [16] analyzed the academic decision-making process in higher education regarding self-selection. Here, students are analyzed in different groups, and their performance is gathered for every activity. In addition to this, individual activities, social responsibility, resource utilization, and social–political context have been investigated. According to these factors, the selection decision is carried out.

## 2. Biographical Data

Biographical data (henceforth bio-data) are information concerning a person's history, that is, about an individual's personality, attitudes, experiences, skills, etc. The data are usually obtained from application forms or questionnaires (biographical inventories), including age, sex, education, work experience, and interests. Bio-data originated from the human resume, a word initially referring to the single sole of shoes, referencing wearing shoes to stepping, so the original meaning of resume is walking and thus, it is interpreted as human experience. The resume was called "foot color" in ancient times and arose in the Sui Dynasty [17].

As a reflection of personal experience, the critical content should be personal experience from birth to resume writing. However, because the ancients attached importance to the influence of the family on personal growth as it truthfully reflected the social status of the leading family and even the individual's family, the resume also wrote the origin, the name of the three generations, and their official position. Therefore, an outline can reflect a person's political attitude, background, social relationship, growth familiarity, and other characteristics. It can become an essential basis for judging their personal quality and political ability [18].

Bio-data began in the late nineteenth century by Peters, who testified at an academic conference that first raised consistent questions about personal life history by noting that life insurance agents could be selected based on candidate answers to a standard set of questions about prior behavior and experience. Since then, various scholars have deliberated and elaborated on the quantitative implication of biographical information projects. In the 1920s, the magnitudes of applied research on bio-data were initiated. Foreign studies have demonstrated that bio-data are an effective and practical instrument for talent expectation and selection [19].

The first research in the field of bio-data in China was conducted in 1992 by Song Guandong in his master's program, titled "The preparation of junior high school teacher

biographical information table". The results showed the retest validity of the "junior high school teacher's biographical information table". It reached 0.84, with an internal consistency reliability $\alpha$ correlation coefficient of 0.80, an internal correlation validity scale above 0.77, and effect correlation validity as high as 0.62. The description scale has significant practical value for predicting and selecting junior high school teachers [20].

Owens and Schoenfeldt, in 1979, conducted a systematic summary with an explanation of bio-data as a life experience through questionnaires. Like a detailed resume information form, the questionnaire contains basic information, education experience, interests, hobbies, family background, health, work experience, life attitude, and values and scores these past events to guess individual compliance, career tendency, academic accomplishment, work performance, and other future performances [21].

In 1991, Mael defined bio-data as "accomplishing the purpose of predicting future work rendering by the magnitudes of past experiences that may affect subject's future behavioral performance" [22]. Mount et al. in 2000 distinguished the research process of bio-data as "facilitating the match between applicants and the selected positions by an understanding of subjects, demographic information, personality characteristics, life experience, and historical information about family background" [23].

By examining the above definitions, some common points were established. Bio-data uses questionnaires as the primary way of information collection. Second, the biographical information questionnaire contains detailed life history information pieces, and lastly, consolidating the biographical information questionnaire project can predict a person's future.

Generally, bio-data present a series of life history information about the target population in a standardized biographical information questionnaire or biographical information blank (BIB) by considering its projects as needed to predict individuals' interests, capabilities, and tendencies to select talents. There are two kinds of modules of the biographical information table. Verifiable modules include sex, age, length of service, education, home address, family background, etc. The other consists of non-verifiable projects, such as a written report, self-work summary, etc.

This work uses Information and Communication Technology (ICT) to collect student biographic data. The main reason for selecting ICT for biographic data is it updates the student information to teachers, scholarship boards, and parents. The recorded student information is easy to update/alter with minimum effort. In addition, the ICT assistive technologies help to track student performance, subject selection, and academic history within a reasonable time. The ICT process integrates with each department that continuously tracks student admission and finance operations with low service time. Frequently assessing the student's learning ability and function helps to understand the student's subject selection efficiency and their impact on learning. Therefore, ICT tools consist of hardware such as tablets, applications, and mobile phones, and software with digital surveys helps to collect student biographic information. The collected digital format information is stored in the database, which allows for analyzing student performance in the education system.

## 3. The Basic Principle of Bio-Data as a Talent Prediction Tool

Even though bio-data are some of the best tools for talent prediction based on the following assumptions, the best predictor of a person's future behavior is what they have done in the past. This hypothesis is proposed precisely based on the rationale of psychometrics and valuable evidence. Some studies show that the average high school score is the best predictor of the average college score. Secondly, asking an individual to describe his previous behavior is more authentic and reliable than discussing his motivation for these behaviors. Lastly, systematic measurements of a person's previous behavior and life experiences can indirectly measure their motivational traits, which may be difficult to measure using other selection methods [24]. Then, the prediction process is defined by using Equation (1)

$$\phi(t+1) = \phi(t-1) + \phi(t-2) - \epsilon(t), \tag{1}$$

In Equation (1), $\epsilon(t)$ is defined as a random factor of; $\rho(\epsilon, \phi) \neq 1$.

### 3.1. The Best Prediction of a Person's Perspective Behavior Is What He Has Already Done

Psychologists have proposed various hypotheses and arguments in bio-data: the developmental integrative model, ecological model, and social identity theory. The developmental integrated model theory was proposed by W. A. Owens (1979) and revealed that "personalities with similar experience in progressive environments are more likely to bring development in life-history conducts in the future." They further stated that different subgroups could be divided according to their life history information to achieve premium predictive results [25]. The ecological model theory was developed by Mumford et al. (1990) and revealed the recursive sequence of individual life events as an acquisition process and used individual differences as one of the predictive factors of future performance [26]. Social identity theory believes that when individuals have a sense of belonging to a new social group, their behaviors may progressively converge toward this social group. More and more social interpersonal relations will share sexual behavior characteristics [27]. For the first time, social identity theory distinguishes the behavioral role between interpersonal and intergroup. It indicates "identity" from particular individual and group levels into certain levels of self-perception: individual identity and a specific social identity. To summarize, social identity theory can be distinct, since all group involvements that individuals participated in the past closely affect their future thinking, patterns, and code of conduct. Thus, social identity theory can be considered a logical supplement and development of the ecological model theory.

### 3.2. Ask Individuals to Describe Their Previous Experiences and Behaviors More Realistically and Reliably Instead of Directly Discussing Motivation

Motivation is an essential psychological concept that refers to the most direct reason behind the incidence of a particular behavior. For example, the learning motivation of a high school student is to obtain relevant college entrance examination results; a murderer's criminal motive is to gain illegal benefits. Motivation is "the internal arousal state that causes and maintains people's behavior in a certain way, mainly manifested as the subjective desire or intention to pursue a certain goal, and people's consciousness for the pursuit of some intended purpose" [28]. Indeed, the motivation behind behavior may be multi-layered and multifaceted. Motives are often examined through questionnaires or interview surveys, where subjects can assess the motivation of their previous behavior. However, due to the bias of the subjects' answers, it is difficult to confirm the authenticity and reliability of their answers for three reasons.

First, subjects are prone to consciously obscure or even falsify reports. Some motives behind some actions involve privacy or departing from the subject's daily routine, causing reluctance to give accurate or complete answers, which can cause partial and imprecise survey results. Second, issues are prone to subjective error and the attribution of one-sided situations. According to Hyde's simple theory of attribution, there is no scientific method of ordinary people's attribution, relying only on their understanding of the world and their introspection to explore the broad motivation. In Kelly's three-degree attribution theory (multiple cue analysis theory/covariant attribution theory), we need to explain behavior in three aspects: to the actor itself, to the objective stimulus (the event or another that prompted the actor to respond to it), and to the situation or relationship where the actor is [25]. It is difficult for ordinary people to do, requiring foremost scholars to explore their motivation in many aspects and levels through scientific techniques. Finally, subjects may not be aware of the essential drivers of current behavioral selection [29]. Human psychological activity is divided into conscious and subconscious, which fails to be detected by us, occurring but not reaching the conscious level. The subconscious has six clinical hypnosis functions: instinct, memory, habits, emotions, energy, and imagination. Although suppressed under consciousness, it always affects people's behavior and decisions [30]. Therefore, the subject individual may be unable to detect the subconscious motivation underlying a specific behavior. If the researcher directly asks the motivation questions, it may be challenging to obtain an accurate answer.

In conclusion, actual and reliable results are not obtained in surveys directly asking motivation class questions. Instead, subjects have detailed and comprehensive descriptions of previous experiences and behaviors, making it easier for us to attribute behavioral motivation in our study precisely. The biography information table dimension covers a wide range, contains virtually all life history events that can determine an individual's current personality and behavior, is highly compatible with motivation and attribution research, and confirms the Jones and Davis correspondent inference theory (1965). The more information a person has, the higher the correspondence of his inferences about behavior [31].

*3.3. Systematic Measurements of a Person's Previous Behavior and Life Experiences Can Indirectly Measure Their Motivational Traits*

Traditional motivation measures mainly comprise three categories: observation measurement method, self-report measurement method, and projection measurement method. Observation measurement refers to the researcher closely observing the subject's behavior to infer the motivation structure. The independent measurement discusses the issue of answering a series of relevant questions through a questionnaire/interview. The researcher analyzes the questionnaire/interview content through scientific methods to deduce the motivation structure, such as by Amabile (1994) [32] and a self-efficacy scale compiled by Schwarzer et al. [33]. Projection measurement is the motivation test without knowing the true purpose of the test. In subsequent analysis, researchers have to extract the subjects' mental state and thinking patterns from the stories woven to assume real motives [34].

The biographical information table (bio-data questionnaire) is a comprehensive survey and collection of subjects' life history information in a questionnaire similar to the motivation scale from the aging measurement. The similarity is that both tested subjects as a questionnaire scale to measure their behavior's motivational characteristics. However, they also differ (as shown in Table 1).

**Table 1.** Comparison between the self-report motivation scale and the biographical information table.

|  | **Self-Report Motivation Scale** | **Biographical Data Questionnaire Scale** |
|---|---|---|
| Measuring dimensions | Single/multi-dimensional | Multi-dimensional |
| Measuring properties | Discuss motivations | Describes behavior |
| Method of scoring | Grade 4–7 score | Basic Information + ratio + score |
| Measurement results | Describe the subject's motivation characteristics | Obtain subject motivation trends |

Single or multi-dimensional subjects characterize the self-report motivation scales. The scales are usually 4–7 rating scales [35] designed to describe the motivation characteristics. The bio-data questionnaire is generally multi-dimensional. The scale can be filled in plus score ratio form with basic information. The motivation trend mainly obtains their measurement through a description of behavior.

The predictive validity of the bio-data questionnaire has performed well in several capacities, especially in terms of performance and talent presentation [36–38]. In addition, its outstanding application field was constantly developed and extended in Table 2 below. Table 2 collates the critical literature on the predictive validity of bio-data in chronological order.

Based on the systematic measurement of previous behavior and life experience, the bio-data questionnaire can be administered to obtain the subjects' motivation characteristics and behavior patterns and generate assignment subscales with good reliability to predict the future behavioral performance of the target group.

**Table 2.** A summary of the predictive validity studies of bio-data.

| Investigator | Publication Year | Study Content and Results |
|---|---|---|
| Ghiselli [39] | 1966 | The system uses the student score performance, test results, and aptitude score values to evaluate the student performance. The predictive validity coefficient of the bio-data was as high as or even higher than 0.5, which was more convincing than most of the evaluation tools at the time. |
| Reilly and Chao [40] | 1982 | This study uses the eight categories to make an effective selection process. During the analysis, the system uses self-assessment, academic achievement, checks, judgment, and technical information to evaluate the student bio-data. Bio-data had an average predictive validity of 0.35 for performance. The study maintains flexibility, reliability, and accuracy while analyzing student bio-data |
| Song Guandong [41] | 1992 1997 | The application of bio-data in selecting junior high school teachers was systematically studied. The study showed that the biography data applies to determining junior high school teachers. The scale's retest dependability and internal consistency reliability were above 0.8, and the validity correlation was 0.62, while reliability and validity reached an equivalent level to foreign studies. The range belongs from 0 to 1. |
| Kilcullen et al. [42] | 1995 | This study intends to construct the temperament by utilizing the lower correlations, temperament measurement, and criteria validity. According to these factors, student performance is predicted with minimum difficulties. The authors found that bio-data have greater incremental validity and effectiveness than traditional personality tests in predicting student performance at school. |
| Karas and West [43] | 1999 | The authors examined the application of bio-data in large-scale recruitment. They showed that a new evaluation system could maximize the fairness and efficacy of the recruitment process to meet customer needs. |
| McManus et al. [44] | 1999 | The authors studied the incremental validity of five prominent personalities to the biographical questionnaire in traditional task performance and extended situation performance in the insurance industry. Later, they found that the four personality dimensions of minor characters can effectively predict the situational performance of insurance salespeople while visibly having predictive validity in task performance. |
| Michael et al. [23] | 2000 | A cross-validation study of 376 office clerical workers showed that biographical measures had more significant incremental variance than the big-five personality structure test (five-factor model; FFM) and the general mental ability test (GMA). |
| Karas and West [45] | 2002 | The authors found that biographical measurement tools can provide higher incremental validity when predicting and evaluating work performance compared to traditional cognitive ability tests. |
| Frederick et al. [9] | 2004 | The system was developed to evaluate college student performance by analyzing their bio-data and performance measures. Their results suggested that bio-data are more effective in predicting academic performance in college students than traditional standardized tests (SAT/ACT). |
| Mael and Hirsch [46] | 2006 | They used two classes of the U.S. Military Academy (2565 people/subjects). Two bio-data collection methods and quasi-rational analysis were applied: the rational scale of bio-data resulting from the existing personality test and rainforest empirical methods to predict their leadership performance. Thus, it indicates that both bio-data measures have higher incremental validity than the college's existing admission scheme, and both showed reasonable stability after cross-validation. |

## 4. Checking College Students' ICT Biographical Information Blank and Its Role in the Selection of Students' Major Courses

The complexity of the socio-division of labor determines the diversity of the majors. For the convenience of research, it is necessary to be divided into several disciplines accordingly. An undergraduate [47] major is commonly divided into twelve fields: science,

engineering, agriculture, economics, medicine, literature, history, philosophy, education, management, law, business, commerce, journalism, and art. Our study was according to the twelve types (or further merged into fewer types) to be involved in the research process.

*4.1. Preparation of the Biographical Information Using ICT Form for College Students*

The preparation process for the college student biographical information table, otherwise known as the bio-data questionnaire, is similar to other selection tests, which can be divided into the following steps: Here, the student biographical information is collected via the ICT assistive technology.

(1) Select the major group within the target population: To compile the biographical information blank for college students, first choose the major type and then locate the target group according to the major type. For example, if you want to fill out the biographical information blank for engineering college students, we select research samples from university engineering students. During this process, the application is created with the help of ICT assistive technology. The application consists of the student's basic information details, preference, location, education subject selection, and major types of categories. These categories are filed by the student and stored in the database to analyze the student's subject preference and performance.

(2) Understand pre-college life history experiences and determine the dimensions of college students' life history information: When the sample is selected, we should consciously pick those typical cases (such as students with particularly outstanding results, mediocre or poor performance via the ICT based devices), and to analyze their university life history experience through in-depth interview, discussion, view experience summary materials, so that we could determine the bio-data information for specific subjects. The ICT applications consist of the student history details, discussions, interview questions, and materials for subject information. The students are instructed to fill in this field in the ICT application, which is used to understand the student history of information.

(3) Prepare a biographical information blank project: According to the biographical information table project of the dimension of life history, the project should reference as much information as possible in the preparation and compile as many biographical information projects as possible, requiring more than 150 projects. Therefore, ICT techniques are incorporated into this education system to collect student information. The gathered information is stored in the database, which is more helpful in understanding the student's biographical details. Once the student enters their preference, it requires prior knowledge and information to alter the existing information. Moreover, the ICT application follows certain policies and procedures while collecting student biographic information.

(4) Select targets: The validity standard is a reference standard to measure the validity of a test. The essential of the reality of bio-data lies in the validity and standard correlation. The digital application-based collected student information correlated with the student performance. A sound effect standard is effective, reliable, and measurable. It shows available data or grade representation, and the measurement method is simple, saving time and effort while being economical and practical. The influential standard commonly used in education measurement includes academic performance and teacher evaluation. According to the conditions and requirements of the effective target, the academic score (grade score) is selected in the study of the college student biographical information table as the adequate standard.

(5) Select project scoring method in the biographical information blank: The biography information table scoring process [48] includes correlation and regression analysis methods; the regression analysis method chooses all ICT bio-data items significantly related to the effect standard and unit weight allocation. Most scholars believe that the regression analysis method is appropriate when there are large samples and fewer items or the correlation between projects is low [49]. The correlation method is the weight analysis method according to the frequency difference of high and low standard group options, dividing subjects into high and low standard groups. The first 50% of issues are high-

standard groups, and the last 50% of subjects are low-standard. Calculate the difference between groups and determine the project weight (score), including horizontal and vertical percentage methods. Table 3 is an example of the flat percentage method.

**Table 3.** Example of the horizontal percentage method.

| Options | High-Standard Group | Low-Standard Group | Total Score | High-Standard Group Percentage | Assigned Weight |
|---|---|---|---|---|---|
| Very fit | 19 | 35 | 54 | 35 | 4 |
| Consistent | 97 | 52 | 149 | 67 | 7 |
| General | 8 | 25 | 33 | 24 | 2 |
| In conformity | 6 | 15 | 21 | 29 | 3 |
| Very inconsistent | 10 | 13 | 23 | 43 | 4 |

(6) Screen the biographical information blank project to form a formal biographical information sheet. Samples are randomly drawn from different subjects, "at least 100 people per subject, if the number of items is more than 100, the number of samples need to be equivalent to the number of items" to perform pre-investigation. After the project score and statistical analysis, Information Communication and Technology-based bio-data questionnaire items are selected to form a formal biographical information table.

(7) Carry out the survey and statistical analysis of the biographical information table and construct a regular model of bio-data: About 500 samples are randomly selected from the student groups of different subjects to carry out the formal questionnaire survey and determine the project score, reliability, and validity test according to various disciplines to construct the regular model of bio-data.

(8) Form a formal biographical information table of college students: The official biographical information table consists of two parts: pre-university life history information and a regular model. The pre-university life history information includes several questions: personal study, hobbies, interests, personality, behavior habits, family environment, parenting methods, etc. The regular model is the realized systematic model.

*4.2. Application of Student Bio-Data in Student Major Selection*

Applying the collected ICT assistive technique bio-data to college student's choice of major enlightens high school students on the college entrance examination. The biographical information blank is filled in to predict or identify students' suitable majors against the bio-data. A student could answer the questions on the biographical information blank and controls. The total score is 356 for the bio-data. Although the total score in other subjects is below 200, the student is suitable for engineering study; sometimes, a high score may be achieved in two or more majors simultaneously so students can refer to their future ambitions in two or more majors. In addition, the ICT-based student response in their major selection and respective performance is illustrated in Table 4.

Table 4 illustrates the ICT-based biographic data collection and student response analysis. Different samples are utilized during the analysis to evaluate the ICT-based student subject selection and performance. Here, different elements such as Benefits of ICT during the lecture (samples = 315), Negative impact of ICT during lecture (samples 315), Impact of ICT on you while another student utilized (samples 259), and Integrating student perception (samples 158) are utilized to system efficiency. In the first criterion, 48.25% of students reported that ICT helps them take notes more easily than the manual process. The analysis was conducted by 49 students, in which 15.5% of students stated that ICT-based developed tools are more useful for acquiring supplementary information. The gathered information allows students to decide while selecting the major subjects in the education system. Then, 25.8% (80) of the responders state that the ICT process used to access the material improves the overall performance system. Likewise, 7.61% (24)

respondents summarized the information, and 3.1% (10) students were engaged with the student used to improve the student perception while utilizing the ICT techniques.

**Table 4.** ICT-based biographic data collection and student response analysis.

| Element | Described Theme | Response (Frequency) (%) |
| --- | --- | --- |
| Benefits of ICT during the lecture (samples = 315) | Note taking easier | 48.25 (152) |
| | Material access | 15.5 (49) |
| | Supplementary information access | 25.8 (80) |
| | Student and instructor engagement | 7.61 (24) |
| | Break from staff | 3.17 (10) |
| The negative impact of ICT during lecture (samples 315) Impact of ICT on you while another student utilized (samples 259) (The result computation for 259) | Interruption | 52.89 (137) |
| | Notes writing with more effective | 9.26 (24) |
| | Distraction | 25.86 (67) |
| | Notes sharing facility | 3.47 (9) |
| | Attention increment | 2.31 (6) |
| | Material understanding | 6.17 (16) |
| Integrating student perception (samples 158) | Interaction improvement | 56.32 (89) |
| | Understanding improvement | 24.0 (38) |
| | Staff engagement | 14.55 (23) |
| | Information summarization | 5.06 (8) |

Thus, according to our previous research on students from engineering majors [50] and students from agricultural majors [51], bio-data have good reliability and validity for predicting students' school performance. The internal consistency of the ICT biographical information table of engineering college students reached 0.871, with the total grade point of college students as the effect standard. The correlation between each item and the validity target reached the significance level. Internal reliability of 0.957, *t*-test sign value of $0.000 < 0.05$, modified card square divided by $2.028 < 3$, R MSEA $0.052 < 0.08$, and IFI and CFI values of $0.902 > 0.9$ showed good fit; NFI and TLI $> 0.8$ showed that the fit was acceptable. Therefore, the revised model fit is still good overall. It shows that college student bio-data can be used as a tool for student's choice of major.

**5. Conclusions**

Bio-data have a history of more than 200 years. The earliest use of bio-data prediction tools concentrates on areas with strong operational skills, such as life insurance agents, store cashiers, bank staff, hotel staff, etc. Studies have shown that ICT-based collected bio-data are better in predicting career proficiency when compared to finger talent, personality, movement, mechanical manipulation, intelligence, and interest tests. Later, bio-data were gradually introduced to work areas with more composite vocational skills, such as in the prediction of civil servants, teachers, and enterprise managers and the prediction of the academic performance of college students. Various results show that biographical information blank items are diverse, and the impurity of the content may lead to low internal reliability ($\alpha$ coefficient is usually 0.60–0.80) but a high test–retest validity coefficient (usually 0–0.90).

In contrast, its validity has predictive validity because it is independent of each score. In addition, because ICT biographical information blanks included verifiable and unverifiable items, the ideal subjects answered more reliably because they were accountable for their responses. Studies show that the description of individual life history was moderately associated with the results recorded by the psychologist. It indicates that the information and communication technology-based gathered bio-data have good accuracy. Our previous

study of bio-data on students from engineering and agricultural majors shows that bio-data can be used for students' major selection. Therefore, our study utilized Information Communication Technology to analyze student performance. The technology-based collected details are investigated according to the student performance and test score values. The effective analysis of bio-data measures leads to maximizing the overall selection process. In future studies, student data have been further utilized to create the training and learning set to improve the automatic subject selection recommendation. In addition, ICT techniques are incorporated with the student bio-data collection process, and a meta-heuristic optimization algorithm is applied to filter the qualified data, which leads to maximizing the overall subject selection efficiency.

**Author Contributions:** Writing of the original draft, preparation, methodology, conceptualization, validation, and formal analysis, Y.W.; supervision, resources, project administration, visualization, and funding acquisition, G.S. All authors have read and agreed to the published version of the manuscript.

**Funding:** This study was funded by the National Social Science Foundation of China, grant number 15BSH093.

**Institutional Review Board Statement:** Not applicable.

**Informed Consent Statement:** Not applicable.

**Data Availability Statement:** Data for this study are available from the corresponding author upon reasonable request.

**Acknowledgments:** We thank "Northeastern University, China" for accompanying us in this research.

**Conflicts of Interest:** The authors declare no conflict of interest.

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
