# Peer review of "Application of Biographical Data in Student’s Major Selection"

_sustainability, doi:10.3390/su142315894_

Round 1
Reviewer 1 Report
The paper addresses an important topic in the area of careers,ICTs and bio strengths. A few elaborations would be helpful to new readers. Thank you.

Reviewer 2 Report
The line of research is clearly identified. Also, the prospect for further research is provided. However, there are some minor points that authors should consider in order to improve their paper. Taking into account the following remarks:
-Table 2 of the manuscript contains little discussion and no contemporary bibliography sources. I think it should be emphasized and a deeper discussion of that part, in particular, should be made.
-There should be more cohesion between the conclusion and the discussion. Also, it could be more organized and structured.
Many of the references are excessively old. For this reason, I consider it can be rewritten by selecting and adding recent sources.
